# Geographical indications and health-conscious behaviors among nursing students: A mixed methods study

Esin Kavuran[1], Erkan Denk [iD][2]*

1 Atatürk University, Faculty of Nursing, Erzurum, Türkiye, 2 Bitlis Eren University, Kanık School of Applied Sciences, Tourism Management, Bitlis, Türkiye

* edenk@beu.edu.tr

## Abstract

This study was conducted using a Sequential Explanatory Design, in which the quantitative phase was carried out first, followed by a qualitative phase to examine the influence of nursing students' attitudes toward purchasing Geographical Indication (GI)–certified products and their mindful eating practices on their healthy eating attitudes. The study population consisted of 1.385 nursing students enrolled in a faculty of nursing. The quantitative component included 392 students who met the inclusion criteria, while the qualitative component comprised 15 students. For data collection, the quantitative phase employed a sociodemographic data form, the Healthy Eating Attitude Scale (HEAS), the Mindful Eating Scale (MES), and the Attitude Scale for Purchasing GI-Certified Products (ASP-GICP). In the qualitative phase, data were collected using a Semi-Structured Interview Form. Quantitative findings showed that younger students (17–25 years) and females had significantly higher scores in GI product attitudes and mindful eating ($p < 0.01$). Students with greater knowledge of GI products and those with healthier personal eating habits demonstrated more positive healthy eating attitudes. Structural Equation Modeling revealed that attitudes toward purchasing GI products (S.P.T = 0.32, t = 3.10, $R^2$ = 0.10) and mindful eating (S.P.T = 0.29, t = 2.90, $R^2$ = 0.08) significantly predicted healthier eating attitudes. Furthermore, mindful eating exhibited a significant moderating effect, strengthening the positive impact of GI product attitudes on healthy eating attitudes ($\beta$ = 0.27, t = 3.10, p = 0.003). Qualitative interviews highlighted the influence of societal and familial norms on students' perceptions of GI products, while cost and limited accessibility emerged as key barriers. The study concludes that integrating GI product awareness and mindful eating practices into nursing education may strengthen students' health-conscious behaviors and support their development as future health professionals.

**Data availability statement:** All relevant data are within the paper.

**Funding:** The author(s) received no specific funding for this work.

**Competing interests:** The authors have declared that no competing interests exist.

## 1. Introduction

According to the World Health Organization, healthy nutrition involves the consumption of adequate amounts of fruits, vegetables, whole grains, proteins, and healthy fats to meet the body's requirements, while limiting the intake of sugar, salt, and processed foods [1]. Geographical Indication (GI) certified products are increasingly preferred by health-conscious consumers. These products are perceived as more natural, less processed, and nutritionally superior due to their being specific to a particular region and produced using traditional, additive-free production methods; this perception strengthens the association of GI products with healthy eating [2]. Moreover, GI certification and official labelling systems particularly the use of recognised logos signal product quality and authenticity, thereby increasing consumer trust; this, in turn, positively influences purchasing decisions and encourages healthier food choices [3].

Worldwide, many countries are recognised for their distinctive GI certified products. France stands out with Camembert cheese (PDO), Germany with Schwarzwalder Schinken, Italy with Prosciutto di Parma, and Spain with Jamón Serrano and Galician Beef [4]. In Asia, India certifies products such as Vripakshi Hill bananas, Kodaikanal Hill garlic, and Madurai Malli jasmine, while Thailand is known for Doi Tung coffee and Thai Hom Mali rice [5]. In Turkey, numerous regional products such as cherries, pomegranates, grapes, beans, and watermelons are protected under the GI system [6].

Within this framework, the growing interest in GI certified products depends not only on the origin and production methods of the products but also on consumers' levels of awareness regarding healthy eating. Indeed, consumers with higher levels of awareness generally prefer fresh, organic, and minimally processed foods and tend to select products with clearly stated nutritional value and low additive content [7–9]. Mindful eating plays an important role in shaping consumer behaviour, particularly within the health and well-being sector; it supports individuals in avoiding highly processed and unhealthy foods while guiding them towards choices aligned with their health goals [9,10].

For this reason, a comprehensive understanding of the factors influencing consumer behaviour is required, highlighting the importance of the Theory of Planned Behaviour (TPB). The TPB defines three fundamental determinants that influence the likelihood of engaging in a particular behaviour: attitudes, subjective norms, and perceived behavioural control [11]. Consumers' positive attitudes towards GI products are largely shaped by perceptions of their health benefits, their contributions to the local economy, and their role in supporting environmental sustainability. Such positive attitudes have been shown to strengthen consumers' intentions to purchase GI products. In addition, favourable opinions expressed by family members, friends, and broader social networks may further increase individuals' tendencies to prefer GI-labelled products [12]. Observing supportive behaviours within the social environment reinforces similar consumption patterns (Fig 1).

It has been found that awareness of GI certification significantly affects consumers' product preferences and willingness to purchase, and that educational level is associated with this awareness [13]. In addition to attitudes, subjective norms, and perceived behavioural control, consumers' behavioural intentions towards GI-labelled products have been shown to be influenced by perceived

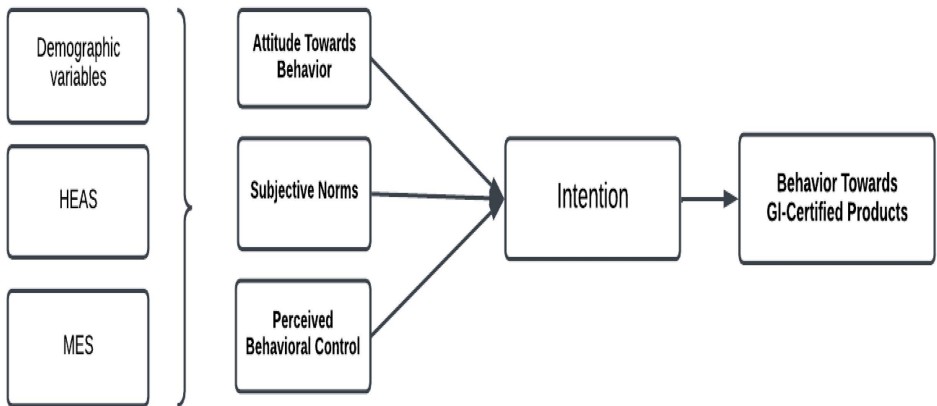

**Fig 1. Consumer behavior toward geographically ındicated products based on the TPB.** HEAS = Healthy Eating Attitude Scale; MES = Mindful Eating Scale.

product value and country-of-origin image [14]. It has also been emphasised that GI certifications enhance perceptions of product quality and that informing consumers plays an important role in increasing the competitiveness of these products [5,15].

As future healthcare professionals, nursing students constitute an important consumer group in terms of health-conscious purchasing behaviour and dietary attitudes [16,17]. It has been demonstrated that higher levels of digital healthy nutrition literacy among nursing students are associated with healthier dietary behaviours [18]. Furthermore, studies examining nursing students' fast-food consumption habits, nutritional knowledge, and attitudes have suggested that nutrition education should be integrated into the curriculum [19]. However, among existing studies, there is no research that directly examines the effect of awareness of GI certified products on nursing students' healthy eating attitudes and purchasing behaviours.

The aim of this study is to examine the effects of nursing students' awareness of and attitudes towards GI certified products on health-conscious behaviours and purchasing preferences, using a mixed-methods approach that combines quantitative and qualitative methods, and to reveal how GI products are perceived within the context of healthy and mindful eating and how these perceptions shape behavioural intentions and actual consumption behaviours.

### Hypotheses

$H_1$: Attitudes toward GI products significantly affect nursing students' healthy eating attitudes.

$H_2$: Mindful eating, which is related to the perceived behavioural control component of the TPB, significantly affects nursing students' healthy eating attitudes.

$H_3$: Attitudes toward GI products and mindful eating together significantly affect nursing students' healthy eating attitudes.

$H_4$: Mindful eating moderates the relationship between attitudes toward GI products and nursing students' healthy eating attitudes.

## 2. Methods

### 2.1. Research design

This study was conducted using a Sequential Explanatory Design, whereby the quantitative phase was undertaken first and followed by a qualitative phase. This approach enabled a comprehensive examination of the influence of nursing

students' attitudes towards purchasing GI certified products and their mindful eating practices on their healthy eating attitudes.

## 2.2. Population and sample

**Quantitative component.** The study population comprised 1.385 students officially registered in the Nursing Faculty of Atatürk University during the 2024–2025 academic year. Random sampling methods were utilised, and the required sample size was calculated using a standard formula for known populations (n = N.Z².p.(1–p)/ [(N–1).E² + Z².p.(1–p)]). Based on a 95% confidence level and a 5% margin of error, the minimum required sample size was determined to be approximately 301 participants.

During the sampling process, a list of enrolled students was obtained from the faculty administration. Although the calculated minimum sample size was 301, a larger sample (n = 400) was selected to ensure adequate representation across all class years and to compensate for potential non-response. Using a stratified random sampling method, a total of 400 students,100 from each class year, were randomly selected and invited to take part in the study. Data collection was completed with 392 students, as eight surveys were excluded due to incomplete responses. The response rate was therefore 98%.

**Qualitative component.** In the qualitative phase of the study, a purposive sampling strategy was employed to select participants. This approach is appropriate when individuals possessing particular characteristics or experiences are required, thereby ensuring the inclusion of participants who are most relevant to the study's aims [20].

A total of 15 participants were included in the qualitative sample. These individuals were deliberately chosen to reflect a range of knowledge, attitudes, and behaviours relating to GI products among nursing students. During recruitment, attention was paid to participants' levels of awareness of GI products, their attitudes towards such products, and their purchasing behaviours. To enhance variation within the sample, efforts were made to ensure a balanced representation across class years and socioeconomic backgrounds.

Participants were selected according to predefined criteria, including prior familiarity with GI products, frequency of consumption, and general health consciousness. This sampling strategy was designed to yield rich and nuanced insights into nursing students' attitudes towards GI products and to contribute to a deeper understanding of consumption behaviours associated with these products [21].

## 2.3. Data collection ınstruments

**Quantitative component.** A socio-demographic data form developed by the researchers was employed in this study. The form comprised items relating to students' demographic characteristics (age, gender, marital status, and place of residence); educational and professional characteristics (class year); family and socio-economic background (household income, presence of a healthcare professional in the family, and family lifestyle and eating habits); personal health and nutrition-related behaviours (self-perceived eating habits, and level of physical activity); and purchasing behaviours concerning GI products, such as frequency of purchase, level of knowledge, and reasons for preference.

*Healthy Eating Attitude Scale (HEAS).* Attitudes towards healthy eating were assessed using the Healthy Eating Attitude Scale (HEAS), developed and validated by Demir and Cicioğlu [22]. The scale consists of 21 items rated on a five-point Likert scale (1 = strongly disagree to 5 = strongly agree) and is structured across four subscales: *nutrition knowledge* (items 1–5), *feelings about nutrition* (items 6–11), *positive nutrition* (items 12–16), and *poor nutrition* (items 17–21). Reported internal consistency coefficients were 0.90, 0.84, 0.75, and 0.83, respectively. Positive items include items 1, 2, 3, 4, 5, 12, 13, 14, 15, and 16, whereas negative items comprise items 6–11, and 17–21 [22]. In the current study, the Cronbach's alpha coefficient was recalculated as 0.86.

*Mindful Eating Scale (MES).* Mindful eating behaviours were measured using the MES-30, originally developed by Framson et al. [23] and adapted into Turkish by Köse et al. [24]. The scale includes seven subscales: *eating without*

*awareness*, *emotional eating*, *eating control*, *mindfulness*, *eating discipline*, *conscious eating*, and *interference* (influence of external factors). Twenty items are reverse scored. Items 1, 7, 9, 11, 13, 15, 18, 24, 25, and 27 are scored positively, whereas the remaining items are reverse scored. Higher subscale scores indicate stronger levels of the respective mindful eating characteristic. The scale also yields a total mindful eating score. The original Cronbach's alpha coefficient was reported as 0.73 [23,24]. In the present study, the reliability coefficient was recalculated as 0.78.

*Attitude Scale for Purchasing GI-Certified Products (ASP-GICP).* Attitudes towards purchasing GI-certified products were measured using the Attitude Scale for Purchasing GI-Certified Products, developed by Yüce and Korucak [25]. The instrument comprises 33 items across three subscales: *knowledge* (items 1–7), *emotional* (items 8–22), and *behavioural* (items 23–33). The original internal consistency coefficient was 0.95. The ASP-GICP is a five-point Likert-type scale with response options ranging from "strongly disagree" (1) to "strongly agree" (5), with higher scores reflecting more positive attitudes towards purchasing GI-certified products [25]. In this study, the Cronbach's alpha coefficient was recalculated as 0.88.

### Qualitative component

*Semi-structured ınterview form.* A semi-structured interview form was developed to obtain an in-depth understanding of nursing students' knowledge, attitudes, mindful eating practices, and healthy eating behaviours in relation to GI products. The development process commenced with a comprehensive review of the literature on GI products, mindful eating, and healthy eating. At this stage, findings and recommendations from previous studies, academic articles, theses, and relevant books were carefully examined to ensure theoretical and empirical grounding.

Open-ended questions were formulated to explore participants' knowledge, attitudes, and behaviours regarding GI products. These items were designed to enable participants to articulate their views freely while ensuring that the discussion remained aligned with the core aims of the study. Flexibility was deliberately built into the question structure to encourage the sharing of thoughts, feelings, and experiences, thereby facilitating richer and more nuanced responses.

The interview form was piloted with 2–3 participants to evaluate clarity and response quality. During this pilot phase, the comprehensibility of the questions and the relevance of participants' responses were assessed, which informed subsequent refinements, including the removal of redundant items and the enhancement of selected questions for greater descriptive clarity. To establish content validity, the draft form was reviewed by academic experts and field specialists, and revisions were made based on their feedback. Furthermore, an internal consistency assessment was undertaken to determine whether the questions effectively measured the intended construct [26].

### 2.4. Data collection

**Quantitative component.** Quantitative data were collected by two trained researchers through face-to-face administration of the survey instruments. Data collection took place in a designated classroom within the Nursing Faculty, and surveys were administered during students' free periods outside scheduled lessons. The data were gathered between March and May 2025. Completion of the data collection instruments required approximately 15–20 minutes per participant. Written informed consent was obtained from all students prior to the administration of the survey and again before the qualitative interviews.

**Qualitative component.** Qualitative data were collected through face-to-face, semi-structured interviews with nursing students. These interviews aimed to explore in depth participants' knowledge and attitudes regarding GI products. Each interview lasted approximately 30–45 minutes, providing sufficient time for participants to articulate their experiences, perspectives, and interpretations in detail.

Interviews were conducted in a quiet, pre-arranged room to ensure a comfortable and distraction-free environment that supported open communication. At the outset, participants were informed about the purpose of the study, and consent was obtained for audio recording. Audio recordings were used to ensure that no information was lost and to facilitate

accurate and detailed analysis. In addition to audio recording, the researcher also employed observational techniques and took field notes throughout the interviews. These notes captured non-verbal cues such as body language, tone of voice, and emotional expressions which contributed to a richer understanding of participants' responses and added depth to the qualitative data.

Following data collection, all audio recordings were transcribed verbatim. The transcripts, along with field notes and observational records, were subsequently analysed to generate further insights and support the interpretation of the qualitative findings [20].

### 2.5. Data analysis

**Quantitative component.** Quantitative data were analysed using SPSS (Statistical Package for the Social Sciences) version 22 and AMOS (Analysis of Moment Structures). Both descriptive statistics and Structural Equation Modelling (SEM) were employed to examine the interrelationships among attitudes towards purchasing GI products, mindful eating, and healthy eating attitudes. Descriptive analyses were conducted to characterise the sociodemographic profile of the participants, including age, gender, educational attainment, and income level. For continuous variables, means and standard deviations were calculated, whereas categorical variables were summarised using frequencies and percentages.

SEM was conducted in two sequential stages. First, the measurement model was assessed to establish the construct validity and reliability of the latent variables. This step ensured that the observed indicators adequately represented the underlying theoretical constructs. Subsequently, the structural model was evaluated to test the hypothesised relationships among the study variables. The analytical framework was informed by the TPB, with particular emphasis on the components of attitudes towards behaviour and perceived behavioural control, which underpin health-related consumption behaviours.

Model fit was evaluated using multiple goodness-of-fit indices recommended in the SEM literature, including the chi-square statistic ($\chi^2$), the Comparative Fit Index (CFI), the Tucker–Lewis Index (TLI), the Root Mean Square Error of Approximation (RMSEA), and the Standardised Root Mean Residual (SRMR). Hypotheses were considered statistically supported when the associated t-values exceeded 1.96, corresponding to a two-tailed significance level of 0.05 [27,28]. In addition, coefficients of determination ($R^2$) were calculated for the endogenous variables to assess the proportion of variance explained by the model, with higher values indicating greater explanatory power.

To examine whether mindful eating moderates the relationship between attitudes toward purchasing GI-certified products and healthy eating attitudes (H4), a moderation analysis was conducted within the SEM framework using composite scale scores. Prior to creating the interaction term, the predictor (ASP-GICP) and moderator (MES) were mean-centered to reduce multicollinearity. An interaction term (ASP-GICP × MES) was then computed using the centered composite scores and included in the model as an exogenous predictor of HEAS. A statistically significant path from the interaction term to HEAS was interpreted as evidence of moderation. The results indicated a significant moderating effect ($\beta = 0.27$, $t = 3.10$, $p = 0.003$), demonstrating that higher levels of mindful eating strengthened the positive relationship between attitudes toward purchasing GI-certified products and healthy eating attitudes [29] (Fig 2).

**Qualitative component.** The qualitative data obtained through semi-structured interviews were analysed using a systematic content analysis approach. The analysis began with the verbatim transcription of audio-recorded interviews, a process essential for ensuring accuracy in capturing participants' responses and enabling rigorous, in-depth interpretation. The transcribed data were subsequently organised into thematic categories derived from participants' narratives. Theme identification represents a crucial stage in qualitative analysis, providing a structured means of interpreting complex data [30].

Emergent themes centred on participants' attitudes towards GI products, mindful eating practices, motivations for purchasing GI products, and perceived barriers to accessing such products. Each theme was elaborated in detail to faithfully reflect participants' experiences and viewpoints. Interpretative analyses were conducted to explore how participants

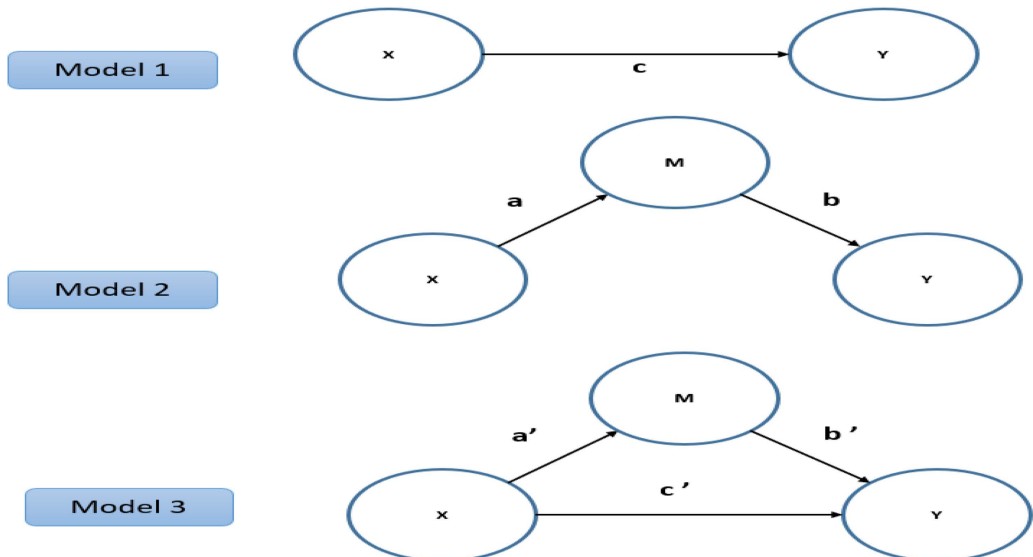

**Fig 2. Structural model testing the moderating role of MES on the relationship between ASP-GICP and HEAS.**

conceptualised their attitudes, shaped mindful eating behaviours, and articulated the factors influencing their purchasing decisions. Challenges encountered in accessing GI products, as well as strategies used to overcome these barriers, were also explored within the thematic framework.

The content analysis process involved identifying recurrent concepts, patterns, and shared experiences embedded within participants' accounts. This systematic approach facilitated the consolidation of key themes and provided deeper insights into collective perspectives. Representative quotations were selected to illustrate each theme and enhance the transparency and credibility of the findings [31].

### 2.6. Ethical statement

The ethics committee permission required for collecting and using the data obtained in this study was obtained from the Atatürk University Non-Interventional Clinical Research Ethics Committee, with decision number 52 in the 6th session held on 27.09.2024.

### 3. Results

#### Quantitative component

The analysis of nursing students' attitudes towards GI products, healthy eating behaviours, and mindful eating practices revealed significant associations with demographic factors, as presented in Table 1. Students aged 17–25 exhibited higher scores on the ASP-GICP (107.5 ± 18.3) and HEAS (63.2 ± 9.8) compared to those aged 26–34 (ASP-GICP: 100.1 ± 19.1; HEAS: 62.7 ± 8.6) (p = 0.001; p = 0.003). Female students scored significantly higher on the ASP-GICP (106.3 ± 18.6) and MES (121.4 ± 15.2) than their male counterparts (ASP-GICP: 103.9 ± 18.9; MES: 118.7 ± 15.4) (p = 0.012; p = 0.018). Single participants demonstrated superior scores in ASP-GICP (105.3 ± 18.5) and HEAS (62.5 ± 10.0) relative to married students (ASP-GICP: 102.0 ± 19.3; HEAS: 61.1 ± 10.7) (p = 0.005; p = 0.008).

Furthermore, students residing in urban areas reported healthier eating attitudes (HEAS: 62.8 ± 10.2) than those from rural regions (HEAS: 62.1 ± 10.3) (p = 0.003). Higher family income was positively correlated with improved ASP-GICP scores, particularly among students from the high-income bracket (106.4 ± 18.6) (p = 0.015). Participants who maintained

**Table 1. Nursing students' attitudes towards HEAS, MES and ASP-GICP based on demographic characteristics.**

| Characteristic | Category | n(%) | ASP-GICP | HEAS | MES |
|---|---|---|---|---|---|
| **Age** | 17-25 | 315(80.4) | 107.5±18.3 | 63.2±9.8 | 122.8±14.9 |
| | 26-34 | 77(19.6) | 100.1±19.1 | 62.7±8.6 | 123.1±15.8 |
| | | | **p=0.001**, t=3.45 | **p=0.003**,t=2.87 | **p=0.002**,t=2.99 |
| **Gender** | Female | 245(62.5) | 106.3±18.6 | 62.9±10.1 | 121.4±15.2 |
| | Male | 147(37.5) | 103.9±18.9 | 61.7±10.4 | 118.7±15.4 |
| | | | **p=0.012**, t=2.54 | p=0.721,t=2.23 | **p=0.018**,t=2.31 |
| **Marital Status** | Married | 5(1.3) | 102.0±19.3 | 61.1±10.7 | 119.3±15.9 |
| | Single | 387(98.7) | 105.3±18.5 | 62.5±10.0 | 120.2±15.1 |
| | | | **p=0.005**, t=2.91 | **p=0.008**, t=2.73 | p=0.506,t=2.82 |
| **Place of Residence** | Rural | 210(53.6) | 104.8±18.8 | 62.1±10.3 | 119.9±15.3 |
| | Urban | 182(46.4) | 105.6±18.4 | 62.8±10.2 | 120.3±15.0 |
| | | | p=0.078, t=2.73 | **p=0.003**, t=2.87 | **p=0.009**,t=2.71 |
| **Class** | 1st Year | 101(25.8) | 106.1±18.7 | 62.6±10.0 | 121.2±15.0 |
| | 2nd Year | 105(26.8) | 105.0±18.9 | 62.3±10.2 | 120.0±15.2 |
| | 3rd Year | 104(26.5) | 104.7±18.6 | 62.0±10.4 | 119.8±15.4 |
| | 4th Year | 82(20.9) | 104.5±19.0 | 62.2±10.1 | 119.7±15.6, |
| | | | p=0.311, t=2.59 | **p=0.016**, t=2.38 | p=0.064,t=2.45 |
| **Family Income** | Low | 151(38.5) | 104.2±18.8 | 61.8±10.5 | 119.5±15.7 |
| | Medium | 140(35.7) | 105.8±18.4 | 62.7±10.1 | 120.6±14.9 |
| | High | 101(25.8) | 106.4±18.6 | 63.1±10.0 | 121.0±15.2 |
| | | | **p=0.015**, t=2.43 | p=0.123, t=2.19 | p=0.620,t=2.26 |
| **Presence of a Healthcare Worker in the Family** | Yes | 56(14.3) | 103.7±19.1 | 61.9±10.4 | 119.4±15.6 |
| | No | 336(85.7) | 105.5±18.5 | 62.6±10.2 | 120.3±15.0 |
| | | | p=0.06, t=2.36 | p=0.225, t=2.16 | p=0.022,t=2.24 |
| **Family's Eating Habits** | Healthy | 123(31.4) | 106.8±18.2 | 63.4±9.9 | 121.7±14.7 |
| | Moderate | 147(37.5) | 105.1±18.9 | 62.2±10.3 | 119.9±15.3 |
| | Unhealthy | 122(31.2) | 103.9±19.0 | 61.6±10.6 | 119.1±15.8 |
| | | | **p=0.010**, t=2.62 | p=0.514, t=2.48 | p=0.512,t=2.57 |
| **Personal Eating Habits** | Healthy | 122(31.2) | 107.2±18.1 | 63.8±8.7 | 122.9±15.3 |
| | Moderate | 138(35.2) | 105.0±18.9 | 63.6±9.7 | 122.1±14.8 |
| | Unhealthy | 132(33.7) | 104.1±18.7 | 62.3±10.2 | 119.9±15.3 |
| | | | **p=0.009**, t=2.67 | p=0.051, t=2.61 | **p=0.010**,t=2.64 |
| **Physical Activity** | High | 174(44.4) | 106.6±18.3 | 63.0±10.0 | 121.5±14.9 |
| | Moderate | 105(26.8) | 104.9±18.8 | 62.1±10.3 | 119.7±15.4 |
| | Low | 113(28.8) | 104.3±18.7 | 61.9±10.4 | 119.5±15.7 |
| | | | p=0.607, t=2.79 | p=0.119, t=2.69 | **p=0.008**,t=2.75 |
| **GI Product Preference Frequency** | Never | 103(26.3) | 104.0±19.0 | 61.8±10.6 | 119.2±15.9 |
| | Rarely | 167(42.6) | 105.5±18.5 | 62.5±10.1 | 120.1±15.2 |
| | Frequently | 122(31.1) | 106.0±18.4 | 62.8±10.0 | 120.5±15.0 |
| | | | **p=0.009**, t=2.70 | p=0.081, t=2.62 | **p=0.010**,t=2.67 |
| **Knowledge of GI Products** | High | 89(22.7) | 107.1±18.2 | 63.5±9.8 | 122.0±14.8 |
| | Medium | 154(39.3) | 105.4±18.7 | 62.4±10.2 | 119.8±15.3 |
| | Low | 149(38.0) | 104.7±18.9 | 62.1±10.4 | 119.6±15.5 |
| | | | **p=0.006**, t=2.85 | **p=0.008**, t=2.77 | **p=0.007**,t=2.82 |

*(Continued)*

**Table 1.** (Continued)

| Characteristic | Category | n(%) | ASP-GICP | HEAS | MES |
|---|---|---|---|---|---|
| **Reasons for Preferring Geographical Indication** | Taste | 121(30.9) | 105.9±18.5 | 62.7±10.1 | 120.4±15.1 |
| | Health | 116(29.6) | 105.2±18.8 | 62.3±10.3 | 119.9±15.3 |
| | Support | 155(39.5) | 105.7±18.4 | 62.6±10.2 | 120.2±15.0 |
| | | | **p=0.008**, t=2.73 | **p=0.010**, t=2.66 | p=0.091,t=2.70 |

**Notes:** HEAS=Healthy Eating Attitude Scale; MES=Mindful Eating Score; ASP-GICP=Attitude towards Selection of Geographically Indicated Certified Products

healthy personal eating habits also scored higher on both the ASP-GICP (107.2±18.1) and HEAS (63.8±8.7) compared to their peers with unhealthy eating patterns (p=0.009).

The frequency of GI product consumption was similarly associated with attitudes, as frequent consumers achieved elevated scores on the ASP-GICP (106.0±18.4) and HEAS (62.8±10.0) (p=0.009). Finally, knowledge of GI products was a significant predictor of students' attitudes, with those demonstrating higher knowledge attaining superior scores on the ASP-GICP (107.1±18.2) and HEAS (63.5±9.8) (p=0.006) (Table 1).

As the attitude towards purchasing GI products increases, it has been observed that healthy eating attitudes also increase (S.P.T=0.32, t=3.10, R²=0.10). Additionally, it has been found that mindful eating also have a significant effect on the healthy eating attitudes of nursing students. As mindful eating increase, it has been determined that students' healthy eating attitudes also improve (S.P.T=0.29, t=2.90, R²=0.08). Furthermore, it has been established that both the attitude towards purchasing GI products and mindful eating significantly affect the healthy eating attitudes of nursing students together. The combination of both attitudes has been shown to positively influence healthy eating attitudes (S.P.T=0.35, t=3.50, R²=0.15) (Table 2).

Mindful eating was found to moderate the effect of attitudes toward purchasing GI products on nursing students' healthy eating attitudes. The analysis results demonstrated a significant moderating effect (Standardized β=0.27, t=3.10, p=0.003). This finding indicated that mindful eating strengthened the positive effect of attitudes toward purchasing GI products on healthy eating attitudes (Table 3).

## Qualitative component

The qualitative phase of this study aimed to explore nursing students' consumer behaviours concerning GI products and to understand how these behaviours relate to attitudes toward healthy eating and mindful eating practices (Table 4).

**Table 2.** The effects of ASP-GICP and MES on nursing students' HEAS.

| Hypothesis | Type of relationship | S.P.T | t | R² | Result |
|---|---|---|---|---|---|
| $H_1$: The attitude towards purchasing GI products has a significant effect on nursing students' healthy eating attitudes. | As the attitude towards purchasing GI products increases, healthy eating attitudes also increase (Linear and Positive Relationship) | 0.32 | 3.10 | 0.10 | Confirmed |
| $H_2$: Mindful eating has a significant effect on nursing students' healthy eating attitudes. | As mindful eating increases, healthy eating attitudes also increase (Linear and Positive Relationship) | 0.29 | 2.90 | 0.08 | Confirmed |
| $H_3$: The attitude towards purchasing GI products and mindful eating together have a significant effect on nursing students' healthy eating attitudes. | Both attitudes positively influence healthy eating attitudes (Linear and Positive Relationship) | 0.35 | 3.50 | 0.15 | Confirmed |

Notes: ASP-GICP=Attitude towards Selection of Geographically Indicated Certified Products; MES=Mindful Eating Score; HEAS=Healthy Eating Attitude Scale

**Table 3. The role of MES on the effect of the ASP-GICP on HEAS.**

| Hypothesis | Standardized β | t | p |
|---|---|---|---|
| H₄: Mindful eating moderates the effect of the attitude towards purchasing GI products on healthy eating attitudes. | 0.27 | 3.10 | 0.003 |

Notes: MES = Mindful Eating Score; ASP-GICP = Attitude towards Selection of Geographically Indicated Certified Products; HEAS = Healthy Eating Attitude Scale

**Table 4. Main themes derived from the qualitative interviews.**

| Theme | Codes | Description | Example quotes |
|---|---|---|---|
| Consumer Behavior Towards GI Products | **Attitude Towards Behavior** | | |
| | Health Benefits Knowledge | Participants' knowledge of the health benefits of GI products and its impact on purchasing decisions | *"When I learn about the story behind GI products, my desire to buy them increases."* (P2) |
| | Positive Attitudes Toward GI Products | Positive thoughts and feelings about GI products | *"As I learn more about GI products, I choose them to enjoy and savor their taste."* (P3) |
| Motivations for Purchasing GI Products | **Subjective Norms** | | |
| | Family Influence | The impact of family members' attitudes and behaviors toward GI products | *"My family prefers local products, so I try to buy them too."* (P5) |
| | Friend Influence | The impact of friends' attitudes and behaviors regarding GI products | *"My friends care about healthy eating, so I also choose GI products."* (P6) |
| | Society's Views | General societal perceptions and attitudes toward GI products | *"GI products are considered healthy in society, which influences my choices."* (P7) |
| Barriers to Purchasing GI Products | **Perceived Behavioral Control** | | |
| | Lack of Information | The effect of insufficient knowledge about GI products | *"I feel like I need to learn more about GI products."* (P9) |
| | Price Barrier | The impact of high prices of GI products on purchasing decisions | *"I love buying local, but sometimes these products are really expensive."* (P10) |
| | Access Barrier | Difficulties encountered in accessing GI products | *"When I want to buy GI products, sometimes not finding them really frustrates me."* (P11) |

P = Participant

## Theme 1: Consumer behaviour towards GI products

Participants' attitudes towards GI products were closely associated with their knowledge of the health benefits of these products. Many students reported that understanding the origin and health advantages of GI products increased their willingness to purchase them. Positive attitudes towards GI products were also reported to enhance overall healthy eating experiences. Illustrative quotes include: *"When I learn about the story behind GI products, my desire to buy them increases."* (P2) *"As I learn more about GI products, I choose them to enjoy and savour their taste."* (P3)

## Theme 2: Motivations for purchasing GI products

Participants' purchasing behaviours were influenced by social factors, including family, peers, and broader societal perceptions. Family preferences often guided students' choices, while peers' health-conscious attitudes shaped their decisions. Additionally, societal recognition of GI products as healthy reinforced participants' purchasing behaviours. Representative quotes include: *"My family prefers local products, so I try to buy them too."* (P5), *"My friends care about healthy eating, so I also choose GI products."* (P6), *"GI products are considered healthy in society, which influences my choices."* (P7)

## Theme 3: Barriers to GI product consumption

Despite generally positive attitudes, participants identified several barriers affecting their ability to purchase GI products. Commonly reported obstacles included insufficient knowledge, high prices, and limited availability. Illustrative quotes

include: *"I feel like I need to learn more about GI products."* (P9).*"I love buying local, but sometimes these products are really expensive."* (P10),*"When I want to buy GI products, sometimes not finding them really frustrates me."* (P11)

## 4. Discussion

In this study, the relationship between nursing students' awareness of GI certified products and their healthy eating attitudes was examined through an integrated analysis of quantitative and qualitative data within the framework of the TPB. The study sample predominantly consisted of single female nursing students aged 17–25 years, a demographic profile consistent with previous research conducted among nursing students [32]. (Mahmoud, 2017). The high proportion of participants from low- and middle-income households, together with the limited presence of healthcare professionals within their families, suggests that health-related knowledge and awareness are largely acquired through formal nursing education. This finding underscores the pivotal role of nursing curricula in fostering nutrition-related knowledge and healthy lifestyle behaviours [33].

An examination of dietary and lifestyle characteristics revealed a relatively balanced distribution of healthy, moderately healthy, and unhealthy eating habits among students and their families, alongside heterogeneous levels of physical activity. These findings are consistent with the literature indicating a gap between nutritional knowledge and daily practices among nursing students, as well as the influence of academic stress and environmental factors on lifestyle [33,34]. The study further demonstrated that students consumed GI-certified products infrequently or not at all and possessed limited knowledge regarding these products. Nevertheless, motivations for preferring GI products such as supporting local producers, sensory qualities, and perceived health benefits suggest that these attitudes are shaped not only by health perceptions but also by social and cultural drivers. Given the scarcity of direct evidence concerning GI products among nursing students, the present study offers an original contribution to the literature, while studies highlighting inconsistencies between food literacy, nutritional knowledge, and consumption behaviours provide a relevant conceptual framework for interpreting the findings [35,36].

Quantitative results indicated that nursing students aged 17–25 years demonstrated higher ASP-GICP and HEAS scores. This finding aligns with previous research suggesting that younger individuals are more receptive to health- and nutrition-related information and can adapt more rapidly to new knowledge [37]. Qualitative interviews further revealed that knowledge acquired during university education enhanced students' awareness in food choices and increased their motivation to adopt healthier dietary practices. However, the literature also reports that younger populations may be at greater risk of eating disorders [38]. This apparent contradiction suggests that nursing education may serve as a protective factor that mitigates potential nutritional risks within this age group.

A further notable quantitative finding was that female students exhibited significantly higher ASP-GICP, MES, and HEAS scores compared with their male counterparts. Qualitative data supported this result, with female participants emphasising a heightened sense of responsibility to act as role models as future health professionals, which in turn shaped their dietary behaviours. This finding is consistent with existing literature indicating greater health and nutrition awareness among women [39,40]. From TPB perspective, more favourable attitudes and stronger subjective norms among female students appear to contribute to healthier eating behaviours.

Students residing in urban areas were also found to have higher HEAS scores. Qualitative findings indicated that these students perceived greater access to healthy food options, GI-certified products, and nutrition-related information. This suggests that urban living environments may enhance perceived behavioural control by facilitating access to resources and information. Previous studies similarly identify place of residence as an influential determinant of dietary behavior [41].

Quantitative analyses demonstrated a strong and positive association between awareness of GI-certified products and healthy eating attitudes. This finding suggests that perceiving GI products as more natural and healthier may promote healthier dietary behaviours. Qualitative data enriched this interpretation, with students describing GI products as "reliable", "additive-free", and "healthier". Participants also reported that knowledge of a product's origin and story increased

their willingness to purchase it. These findings are consistent with literature indicating that product knowledge strengthens positive attitudes and purchase intentions [42,43].

According to TPB, attitudes represent one of the strongest predictors of behavior [44]. In this study, positive attitudes towards GI products were associated with enhanced healthy eating behaviours, thereby supporting this theoretical framework. Similarly, previous studies have shown that favourable attitudes towards organic and local foods increase both healthy eating behaviours and purchase intentions [45].

Quantitative findings further revealed that family and friends' healthy eating preferences significantly influenced students' decisions to purchase GI-certified products. This social influence was clearly illustrated in qualitative interviews, as exemplified by a participant's statement: "My family prefers local products, so I try to buy them as well." These findings align with existing evidence highlighting the critical role of social environments in shaping dietary behaviours [46,47]. Within the TPB framework, strong subjective norms appear to facilitate the adoption of healthier eating practices.

With regard to perceived behavioural control, quantitative data indicated that students who felt confident in accessing and purchasing GI-certified products exhibited healthier eating attitudes. However, qualitative findings identified lack of knowledge and higher prices as key barriers limiting GI product consumption. Statements such as "I need to learn more about GI products" highlight the clear need for educational interventions. These findings are consistent with prior research identifying low awareness and economic constraints as major barriers to GI product consumption [48].

Another noteworthy finding of the study is the moderating role of mindful eating in the relationship between attitudes toward purchasing GI-certified products and healthy eating attitudes. Quantitative analyses indicated a significant interaction effect, showing that higher levels of mindful eating strengthened the positive association between attitudes toward purchasing GI-certified products and healthy eating attitudes. Qualitative data further supported this result, with students emphasising that "knowing what you eat" and "paying attention to food labels" directly influence food choices and make healthier decisions more likely. These findings are consistent with previous studies demonstrating that mindful eating promotes healthier eating behaviours [49,50].

Overall, when quantitative and qualitative findings are considered together, this study demonstrates that awareness of GI-certified products, mindful eating, and healthy eating behaviours form a mutually reinforcing structure among nursing students. Within the framework of the TPB, positive attitudes, supportive subjective norms, and higher perceived behavioural control emerge as key determinants of health-conscious eating behaviours. In this context, nursing education should be regarded as a critical factor that simultaneously strengthens all three components.

## 5. Limitations

This study has several limitations that should be addressed. First, the cross-sectional design limits the ability to draw causal inferences regarding the relationships between GI product awareness, mindful eating, and healthy eating behaviors. Longitudinal studies are necessary to explore these relationships more robustly over time. Second, the sample is restricted to nursing students from a single university, which may affect the generalizability of the findings to other populations or regions. Future studies should include more diverse samples across different educational and cultural settings. Additionally, the use of self-reported data introduces the potential for social desirability bias, as participants may tend to overreport their healthy behaviors and mindful eating practices.

## 6. Implications for practice and education

The findings of this study emphasise the importance of enhancing awareness of healthy products and health-conscious eating behaviours among university students, particularly within health-related educational contexts. Extending such initiatives to students enrolled in disciplines such as medicine, dietetics, and public health may help to identify whether similar behavioural patterns exist across different professional groups and may further support interdisciplinary approaches to

nutrition education. Moreover, addressing structural barriers, including cost and limited accessibility, appears essential for facilitating the wider adoption of healthy products among student populations.

Educational strategies that integrate theoretical instruction with experiential learning are likely to be particularly effective. Organising educational visits to healthy product production centres, such as organic farms or traditional food producers, may enhance students' understanding of production processes, product authenticity, and associated health benefits. These experiential learning opportunities have the potential to strengthen food literacy and promote more informed and reflective consumption decisions.

Furthermore, embedding content related to healthy eating and healthy products within university curricula especially in health sciences programmes may improve students' capacity to make evidence-based dietary choices and to deliver accurate and credible nutrition guidance in their future professional roles. Complementary approaches, including social media–based campaigns, interactive educational activities, and student-centred initiatives such as cooking competitions or local food festivals, may further increase student engagement and positively influence both purchasing intentions and consumption behaviours.

## Conclusion

In conclusion, this mixed methods study demonstrates that awareness of healthy products, mindful eating, and healthy eating behaviours are closely interconnected among nursing students. When interpreted through the lens of the TPB, positive attitudes, supportive subjective norms, and higher perceived behavioural control emerge as key determinants of health-conscious eating behaviours. Collectively, these findings highlight the critical role of nursing education in simultaneously strengthening these components and in supporting the development of future health professionals who are well equipped to model and promote healthy dietary practices.

## Acknowledgments

We would like to thank nursing students who agreed to be interviewed in this study and answered the questions sincerely.

## Author contributions

**Conceptualization:** Erkan Denk.

**Formal analysis:** Esin Kavuran.

**Methodology:** Esin Kavuran.

**Resources:** Erkan Denk.

**Software:** Erkan Denk.

**Supervision:** Esin Kavuran, Erkan Denk.

**Writing – original draft:** Esin Kavuran.

**Writing – review & editing:** Esin Kavuran, Erkan Denk.

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
