## [Decision Letter · Decision Letter 0]

10 Dec 2025

Dear Dr. DENK,

Thank you for submitting your manuscript to PLOS ONE. After careful consideration, we feel that it has merit but does not fully meet PLOS ONE’s publication criteria as it currently stands. Therefore, we invite you to submit a revised version of the manuscript that addresses the points raised during the review process.

We look forward to receiving your revised manuscript.

Kind regards,

Hansani Madushika Abeywickrama, Ph.D.

Academic Editor

PLOS One

**Journal Requirements:**

2. We note that your Data Availability Statement is currently as follows:

“All relevant data are within the manuscript and its Supporting Information files.”

Reviewers' comments:

Reviewer's Responses to Questions

**Comments to the Author**

1. Is the manuscript technically sound, and do the data support the conclusions?

Reviewer #1: Yes

Reviewer #2: Yes

Reviewer #3: Partly

Reviewer #4: No

Reviewer #5: Yes

2. Has the statistical analysis been performed appropriately and rigorously?

Reviewer #1: Yes

Reviewer #2: Yes

Reviewer #3: I Don't Know

Reviewer #4: No

Reviewer #5: Yes

3. Have the authors made all data underlying the findings in their manuscript fully available?

Reviewer #1: Yes

Reviewer #2: Yes

Reviewer #3: No

Reviewer #4: No

Reviewer #5: Yes

4. Is the manuscript presented in an intelligible fashion and written in standard English?

Reviewer #1: Yes

Reviewer #2: Yes

Reviewer #3: No

Reviewer #4: No

Reviewer #5: Yes

Reviewer #1: Dear Authors,

The manuscript investigates the relationship between nursing students’ attitudes toward Geographical Indication (GI) products, mindful eating, and healthy eating behaviors, using a mixed-methods approach grounded in the Theory of Planned Behavior (TPB). The topic is relevant for public health, consumer behavior, and nutrition research, and falls within the scope of PLOS ONE. The study combines quantitative (SEM, moderation analysis) and qualitative (semi-structured interviews) methods, offering complementary insights.

The manuscript presents a valuable contribution, but several methodological, conceptual, and structural issues require attention before the study can be considered for publication. The manuscript presents:

- While GI products are an emerging topic in consumer studies, the manuscript should better demonstrate the specific gap it fills, especially in relation to previous studies linking GI products and health-related behaviors.

-The literature review should more clearly articulate why nursing students constitute a unique population for investigating GI-related behaviors.

Specifically:

1. Explicitly define and justify the mixed-methods design; cite methodological authorities (e.g., Creswell, Tashakkori & Teddlie).

2. Strengthen the rationale for studying GI-related attitudes in nursing students and demonstrate the gap in the literature more effectively.

3. Provide more detail on SEM model diagnostics and assumptions.

4. Clarify qualitative analytic procedures, including coding, theme development, reliability, and saturation.

5. Remove causal language and adjust interpretations to reflect cross-sectional associations.

6. Revise sections that offer prescriptive recommendations to avoid overstated implications.

7. Edit the manuscript thoroughly for English grammar, clarity, and academic tone.

8. Ensure consistent referencing style according to PLOS ONE formatting.

Reviewer #2: The article just has minor corrections for spelling and consistency on capitol letters for headings. I think although the theories contributes to the discussions, the article is a bit confusing and maybe should two articles be considered.

Reviewer #3: The study addresses a very interesting topic but the manuscript needs considerable editing. All sections need to be revised (details added to the manuscript as track changes) and some English language editing is required. There is no statement about data availability.

Reviewer #4: The study investigates an important and understudied topic: the role of Geographical Indication (GI) product awareness in shaping nursing students’ healthy eating attitudes. The mixed-methods design strengthens the contribution, and the manuscript is well-motivated using the Theory of Planned Behavior (TPB). However, several methodological, analytical, and reporting issues need to be addressed before the manuscript can be considered for publication.

Reviewer #5: Reviewer's comments:

• In the introduction give examples of the Geographical Indication (GI) certified products from different counries or cultures.

• Under each graph/chart write the full name of the abbreviations, and symbols.

• Please review and clarify the sentence in line 177 and 178 because "relationship between GI product purchasing attitudes and healthy eating attitudes (H1) and between healthy eating attitudes (H2)".

• 2.1.2. Population and Sample

• In Line 196 "1385 students enrolled in the Nursing Faculty", not clear. Does it mean that the sample was taken from faculty who were previously students in the same university?. or this is the name of the nursing school?

• The explanation of teh sampling seems like stratefied sampling but it needs to be mentioned clearly in the methods.

• In line 224 , you need to mention the type of reliability coefficient. Is that Cronbach alpha or another reliability measure.

• The heading 2.1.4. Data Collection is repeated twice in line 244 and 245

• Provide a reference supporting this statement "Hypotheses were considered supported if the t-value was greater than 1.96 ".

• In Structural Equation Modeling (SEM), fit indices are sused as tatistics that assess how well the theoretical model matches the observed data, with key ones including Chi-Square 2), RMSEA (Root Mean Square Error of Approximation), CFI (Comparative Fit Index), TLI (Tucker-Lewis Index), and SRMR (Standardized Root Mean Residual). At least some of them should be measured and added to support the validity of the suggested models.

• In the dada analysis of quantitative data, Using R alone is not sufficient in judging the hypothetical models. There are some

• In qualitative data collection: no mention about reaching saturartion in data collection. It needs to be addressed.

**Do you want your identity to be public for this peer review?** For information about this choice, including consent withdrawal, please see our Privacy Policy

Reviewer #1: No

Reviewer #2: No

Reviewer #3: **Yes:** Mona Al Nsour RN, PhD

Reviewer #4: No

Reviewer #5: No

---

## [Author Response · Author response to Decision Letter 1]

19 Jan 2026

Reviewer's comments:

We would like to thank the reviewers for their thoughtful and constructive feedback. Their suggestions were highly relevant and helped improve the overall structure, clarity, and scholarly contribution of the manuscript. All recommended revisions have been implemented accordingly.

Reviewer 1

Response

In the introduction give examples of the Geographical Indication (GI) certified products from different counries or cultures.

Geographical Indication (GI) certified products are becoming an important choice for health-conscious consumers. These products are associated with specific regions, produced using traditional methods, and are perceived as more natural, less processed, and nutritionally superior due to their characteristics (Belletti et al., 2017). For instance, different countries have various GI-certified products: France has Camembert cheese (PDO), Germany is known for Schwarzwalder Schinken, Italy produces Prosciutto di Parma, Spain has Jamón Serrano and Galician Beef, India certifies Vripakshi Hill bananas, Kodaikanal Hill garlic, and Madurai Malli jasmine, Thailand offers Doi Tung coffee and Thai Hom Mali rice, and Turkey protects products such as cherries, pomegranates, grapes, beans, and watermelons (references). Studies show that GI-certified products are considered healthier due to their pure ingredients and traditional production processes (Fernández-Zarza et al., 2021; Török et al., 2020).

Response

Under each graph/chart write the full name of the abbreviations, and symbols.

Table 1. Nursing Students' Attitudes Towards HEAS, MES and ASP-GICP Based on Demographic Characteristics

Notes: HEAS = Healthy Eating Attitude Scale; MES = Mindful Eating Score; ASP-GICP = Attitude towards Selection of Geographically Indicated Certified Products

Table 2. The Effects of ASP-GICP and MES on Nursing Students' HEAS

Notes: ASP-GICP=Attitude towards Selection of Geographically Indicated Certified Products; MES=Mindful Eating Score; HEAS=Healthy Eating Attitude Scale

Table 3. The Role of MES on the Effect of the ASP-GICP on HEAS

Notes: MES=Mindful Eating Score; ASP-GICP=Attitude towards Selection of Geographically Indicated Certified Products; HEAS=Healthy Eating Attitude Scale

Response

Please review and clarify the sentence in line 177 and 178 because "relationship between GI product purchasing attitudes and healthy eating attitudes (H1) and between healthy eating attitudes (H2)".

In the first step, the aim was to determine whether there is a significant direct relationship between GI product purchasing attitudes and healthy eating attitudes (H1), and whether healthy eating attitudes significantly influence GI product purchasing behavior (H2)

Response

2.1.2. Population and Sample

In Line 196 "1385 students enrolled in the Nursing Faculty", not clear. Does it mean that the sample was taken from faculty who were previously students in the same university?. or this is the name of the nursing school? The study population consisted of 1.385 students who were officially registered in the Nursing Faculty of Atatürk University during the 2023–2024 academic year.

Response

The explanation of teh sampling seems like stratefied sampling but it needs to be mentioned clearly in the methods.

During the sampling process, a list of students enrolled in the 2023–2024 academic year was obtained from the faculty administration. Using a stratified random sampling method, 100 students were randomly selected from each class year, resulting in a total of 400 students invited to participate in the study. The study was completed with 392 students, as 8 participants did not fully complete the survey.

Response

In line 224 , you need to mention the type of reliability coefficient. Is that Cronbach alpha or another reliability measure. In this study, the Cronbach's alpha reliability coefficient of the scale was recalculated as 0.86.

Response

2.1.4. Data Collection is repeated twice in line 244 and 245

Thank you for your valuable comment. However, we respectfully would like to clarify that the heading “2.1.4. Data Collection” appears only once in the revised manuscript. We have carefully rechecked the document, and no repetition of this heading was detected in lines 244–245.

Response

Provide a reference supporting this statement "Hypotheses were considered supported if the t-value was greater than 1.96 ".

The hypotheses were considered supported when the t-value exceeded 1.96, which corresponds to the critical value for a two-tailed test at the 0.05 significance level (Field, 2018; Hair et al., 2019).

Field, A. (2024). Discovering statistics using IBM SPSS statistics. Sage publications limited.

Hair, J. F. (2009). Multivariate data analysis.

Response

In Structural Equation Modeling (SEM), fit indices are sused as tatistics that assess how well the theoretical model matches the observed data, with key ones including Chi-Square 𝜒2), RMSEA (Root Mean Square Error of Approximation), CFI (Comparative Fit Index), TLI (Tucker-Lewis Index), and SRMR (Standardized Root Mean Residual). At least some of them should be measured and added to support the validity of the suggested models.

In the dada analysis of quantitative data, Using R alone is not sufficient in judging the hypothetical models. There are some

In qualitative data collection: no mention about reaching saturartion in data collection. It needs to be addressed.

The quantitative data collected from the study were analyzed using SPSS (Statistical Package for the Social Sciences) version 22 and AMOS (Analysis of Moment Structures) software. A combination of descriptive statistics and Structural Equation Modeling (SEM) was used to examine the relationships among the variables, such as geographical indication (GI) product purchasing, mindful eating, and healthy eating attitudes. Descriptive statistics were applied to summarize participants’ sociodemographic characteristics (e.g., age, gender, education level, income). Measures of central tendency (mean, median) and variability (standard deviation, range) were calculated for continuous variables, whereas frequency distributions were used for categorical variables. SEM analysis was conducted in two stages. First, the measurement model was evaluated to verify the validity and reliability of the latent constructs. Second, the structural model was tested to assess the hypothesized relationships among variables. Model adequacy was examined using commonly recommended fit indices, including the Chi-square (χ²) statistic, the Comparative Fit Index (CFI), the Tucker–Lewis Index (TLI), the Root Mean Square Error of Approximation (RMSEA), and the Standardized Root Mean Residual (SRMR). These indices were used to determine whether the proposed model demonstrated an acceptable level of fit to the observed data. The hypotheses were considered supported when the t-value exceeded 1.96, which corresponds to the critical value for a two-tailed test at the 0.05 significance level (Field, 2018; Hair et al., 2019). The R² values were calculated for the dependent variables (healthy eating attitudes) to indicate the proportion of variance explained by the independent variables (GI product purchasing, mindful eating, and food awareness). Higher R² values indicate stronger explanatory power of the model. In addition, quantitative model evaluation was not based on R software alone; instead, SEM-based fit indices and parameter estimates were used to ensure a comprehensive assessment of the proposed hypothetical models, as recommended in the SEM literature.

Reviewer 2

Response

Consider shorter focused title ‘Geographical Indications and Health-Conscious Behaviors Among Nursing Students: A Mixed Methods Study Geographical Indications and Health-Conscious Behaviors Among Nursing Students: A Mixed Methods Study

Response

Include specific results e.g., percentages or statistical significance

This study examined the relationship between nursing students' awareness of Geographically Indicated (GI) certified products, mindful eating behaviors, and healthy eating attitudes. Using the Theory of Planned Behavior as the framework, it was hypothesized that positive attitudes toward GI products, shaped by social norms and perceived behavioral control, would enhance students' healthy eating behaviors. A cross-sectional study was conducted with 392 nursing students from a faculty population of 1,385. Data were collected using scales measuring healthy eating attitudes, mindful eating, and attitudes toward purchasing GI-certified products. Quantitative findings showed that younger students (17–25 years) and females had significantly higher scores in GI product attitudes and mindful eating (p<0.01). Students with greater knowledge of GI products and those with healthier personal eating habits demonstrated more positive healthy eating attitudes. Structural Equation Modeling revealed that attitudes toward purchasing GI products (S.P.T=0.32, t=3.10, R²=0.10) and mindful eating (S.P.T=0.29, t=2.90, R²=0.08) significantly predicted healthier eating attitudes. Furthermore, mindful eating exhibited a significant moderating effect, strengthening the positive impact of GI product attitudes on healthy eating attitudes (β=0.27, t=3.10, p=0.003). Qualitative interviews highlighted the influence of societal and familial norms on students’ perceptions of GI products, while cost and limited accessibility emerged as key barriers. The study concludes that integrating GI product awareness and mindful eating practices into nursing education may strengthen students’ health-conscious behaviors and support their development as future health professionals.

Response

Over all the manuscript needs editing for language.

The manuscript has been thoroughly revised for English language, grammar, and clarity throughout the text.

Response

The introduction should be more brief and to the point, include recent literature.

The Introduction has been revised to be more concise, and recent literature has been included.

Response

Consider removing the hypotheses...

We appreciate this suggestion; however, the hypotheses were retained to preserve the theoretical coherence of the study and to clearly link the study aims with the analytical framework.

Response

The methods section need to be reorganized to reflect one study. The findings should only report the findings and the discussion section needs to be revised. Findings should be discussed within the context of the literature. A brief conclusion would be beneficial. The reference style should be according to journal requirements. The manuscript has been thoroughly revised accordingly: the Methods section was reorganized, Findings and Discussion were clearly separated, findings were discussed in relation to the literature, a brief Conclusion was added, and references were formatted according to journal guidelines.

Response

Cite WHO correctly

The WHO citation has been corrected and is now cited appropriately as (WHO, 2022).

Response

Lines 51–55 repetitive

Repetition in lines 51–55 was removed and the text was revised accordingly.

Certification and labeling of GI products, including the use of official logos, enhance consumer confidence by signaling quality and authenticity, thereby influencing purchasing decisions and supporting healthier food choices (Calboli, 2015; Vandecandelaere et al., 2021).

Response

This section on conceptual framework is not necessary. I suggest you Delete it The conceptual framework section was removed.

Response

Are you using conscious eating practices interchangeably with mindful eating ? This is confusing…. Terminology has been standardized to “mindful eating” throughout the manuscript..

Response

Already used abbreviation before...continue using abbreviation Abbreviation use has been standardized throughout the manuscript.

Response

Clarify what HEAS and MES stand for in the figure

Figure 1. Consumer Behavior Toward Geographically Indicated Products Based on the Theory of Planned Behavior. HEAS = Healthy Eating Attitude Scale; MES = Mindful Eating Scale.

Response

Either stick to TPB abbreviation or write out theory of planned behavior each time….don’t shift back and forth Usage has been standardized to “TPB” throughout the manuscript.

Response

Do not interpret findings in this section Interpretation was removed from this section.

Response

Be consistent with the use of tenses, use passive voice and make more direct statement of the findings. Tense usage was standardized to past tense and passive voice.

Response

From line 41 to 185 reorganize the literature so it all comes under the Introduction heading. Currently there are a lot of redundancies and there is no need to explain the theory of planned behavior per se. instead use it to organize the literature and then provide a clearer rationale for the study and why the relationship of GI products and health behaviors particularly are important for nursing students. The literature was reorganized under the Introduction heading, redundancies were removed, and TPB was used to structure the literature rather than being explained in detail. The study rationale was clarified accordingly.

Response

Move section on how analysis was done to the analysis subsection under Materials and Methods of the manuscript. the analysis of data is typically included in the methodology section, where researchers outline their analytical strategies and justify their choice of methods. When you rewrite this section brake down the analysis steps more explicitly to make it easier for readers to follow. The analysis section was relocated to Materials and Methods and rewritten with clearer

Response

Methods: I suggest you combine study 1 and study 2 when you write this section. This is Mixed methods research (Sequential Explanatory Design) and then you can subdivide to 1. Quantitative component and 2. qualitative component instead of study 1 and study 2 The Methods section was reorganized as a mixed methods study (Sequential Explanatory Design), with quantitative and qualitative components presented as subsections.

Response

Include a brief justification for why you chose a sample size of 400, even though the calculated sample size was 301. And State the response rate

During the sampling process, a list of students enrolled in the 2023–2024 academic year was obtained from the faculty administration. Although the minimum required sample size was 301, a larger sample (n=400) was selected to ensure adequate representation of each class year and to account for possible non-response. Using a stratified random sampling method, a total of 400 students 100 from each class year were randomly selected and invited to participate in the study. The study was completed with 392 students, as 8 students did not fully complete the survey. The response rate was 98%.

Response

If you calculated the reliability coefficient in your study you should remove the reference from here...I think it should precede this sentence. The reference was repositioned accordingly.

Response

Specify how many interviewers collected the data and if they had training... The data were collected by two researchers through face-to-face interviews

Response

The use of hypotheses in this qualitative part of the study is not appropriate...please remove them and write emeging themes only. Hypotheses were removed from the qualitative section, and emerging themes are presented instead.

Response

Too long paragraph, divide it

The paragraph was divided into three paragraphs.

Semi-Structured Interview Form

In this study, a semi-structured interview form was developed to deeply understand nursing students' knowledge, attitudes, mindful eating practices, and healthy eating behaviors related to geographical indication (GI) products. The development process began with a comprehensive review of the existing literature on GI products, mindful eating, and healthy eating. In this stage, findings and recommendations from previous studies, academic articles, theses, and relevant books were taken into consideration.

Open-ended questions were prepared to explore participants’ knowledge, attitudes, and behaviors regarding GI products. These questions were designed to allow participa

---

## [Decision Letter · Decision Letter 1]

17 Feb 2026

Geographical Indications and Health-Conscious Behaviors Among Nursing Students: A Mixed Methods Study

PONE-D-25-55641R1

Dear Dr. DENK,

We’re pleased to inform you that your manuscript has been judged scientifically suitable for publication and will be formally accepted for publication once it meets all outstanding technical requirements.

Kind regards,

Hansani Madushika Abeywickrama, Ph.D.

Academic Editor

PLOS One

Additional Editor Comments (optional):

Reviewers' comments:

Reviewer's Responses to Questions

**Comments to the Author**

Reviewer #2: All comments have been addressed

Reviewer #3: All comments have been addressed

Reviewer #4: All comments have been addressed

Reviewer #5: All comments have been addressed

2. Is the manuscript technically sound, and do the data support the conclusions?

Reviewer #2: Yes

Reviewer #3: Yes

Reviewer #4: Yes

Reviewer #5: Yes

3. Has the statistical analysis been performed appropriately and rigorously?

Reviewer #2: Yes

Reviewer #3: Yes

Reviewer #4: Yes

Reviewer #5: Yes

4. Have the authors made all data underlying the findings in their manuscript fully available?

Reviewer #2: Yes

Reviewer #3: No

Reviewer #4: Yes

Reviewer #5: Yes

5. Is the manuscript presented in an intelligible fashion and written in standard English?

Reviewer #2: Yes

Reviewer #3: Yes

Reviewer #4: Yes

Reviewer #5: Yes

Reviewer #2: I am satisfied for the responses given to the previous comments and suggestions by the reviewers. The authors addresses the technical aspects, clearly indicated the significant outcomes in the results chapter. The limitations of the study was also included which is important for future research.

Reviewer #3: I would like to thank the authors for their thorough responses to the reviewer comments, the revision has significantly improved the quality of the manuscript.

Reviewer #4: This revised manuscript provides a clear and well-structured examination of the relationship between Geographical Indications, mindful eating, and health-conscious behaviors among nursing students using a mixed-methods approach. The authors have effectively addressed the concerns raised in the previous review, and the study is methodologically sound. The quantitative methodology, including sample size and Structural Equation Modeling, is appropriate, and the conclusions are supported by the data. The qualitative component is well integrated, providing meaningful context to the quantitative findings, and data collection and analysis procedures are adequately described.

The manuscript is generally well-written, with improved tables, figures, and logical organization. Minor language or stylistic refinements could further enhance readability, and a brief clarification confirming that all underlying quantitative datasets are included would strengthen compliance with PLOS ONE’s data-sharing requirements.

Generally, the study meets the journal’s criteria for publication as a technically sound investigation, and the submission is well-positioned for acceptance.

Reviewer #5: (No Response)

**Do you want your identity to be public for this peer review?** For information about this choice, including consent withdrawal, please see our Privacy Policy

Reviewer #2: No

Reviewer #3: **Yes:** Dr Mona Al Nsour

Reviewer #4: No

Reviewer #5: **Yes:** Dr. Zainab Ambani

---

## [Editor Report · Acceptance letter]

PONE-D-25-55641R1

PLOS One

Dear Dr. DENK,

I'm pleased to inform you that your manuscript has been deemed suitable for publication in PLOS One. Congratulations! Your manuscript is now being handed over to our production team.

Kind regards,

on behalf of

Dr. Hansani Madushika Abeywickrama

Academic Editor

PLOS One